# Related Factors with Depression and Anxiety in Mastectomized Women Breast Cancer Survivors

**DOI:** 10.3390/ijerph20042881

**Published:** 2023-02-07

**Authors:** Sergio Álvarez-Pardo, José Antonio de Paz, Ena Montserrat Romero-Pérez, José Manuel Tánori-Tapia, Pablo Alejandro Rendón-Delcid, Jerónimo J. González-Bernal, Jessica Fernández-Solana, Lucía Simón-Vicente, Juan Mielgo-Ayuso, Josefa González-Santos

**Affiliations:** 1Department of Sports, Isabel I University, 09003 Burgos, Spain; 2Institute of Biomedicine (IBIOMED), University of León, 24071 León, Spain; 3Division of Biological Sciences and Health, University of Sonora, Hermosillo 83000, Mexico; 4Department of Health Sciences, University of Burgos, 09001 Burgos, Spain

**Keywords:** breast cancer, mastectomy, depression, anxiety, HADS

## Abstract

Breast cancer (BC) is the most common cancer diagnosis with the highest mortality rate worldwide. The aim of this study was to identify factors related to depression and anxiety in mastectomized women BC survivors. A cross-sectional study was conducted with a sample of 198 women diagnosed with BC aged 30–80 years in Mexico. Depression and anxiety were assessed using the 14-item Hospital Anxiety and Depression Scale (HADS). The results showed that 94.44% and 69.18% of the women scored more than eight points on HADS in the anxiety and depression subscales, respectively; 70.20% and 10.60% were identified as pathological. The following variables were analyzed: age, time elapsed since the start of treatment, received treatment at the time of the evaluation, type of surgery, family history, marital status and employment status. Time elapsed since surgery, having a partner, and employment showed significant results as factors associated to levels of depression and anxiety in these patients. In conclusion, it has been shown that BCSs under 50 years of age receiving some kind of treatment, without family history, without a partner, with a job, with more than secondary education and with more than 5 years since diagnosis could have higher rates of clinical depression. On the other hand, BCSs older than 50 years receiving some kind of treatment, without family history, without a partner, with a job, with more than secondary education and with more than 5 years since diagnosis, could have higher rates of clinical anxiety. In conclusion, the variables studied provide valuable information for the implementation of psychotherapy plans in healthcare systems to reduce the risk of depression and/or anxiety in women with BC who have undergone mastectomy.

## 1. Introduction

Breast cancer (BC) is the most common cancer diagnosis among women worldwide and has the highest mortality rate [1,2], with the survival rate being close to 80% in European countries [3]. In Mexico, BC has also been the most common cancer diagnosed in the female population since 2006 [4]. Only 10% of all cancers are diagnosed at an early stage, so the risk of dying as a result of BC is doubled in Latin America in contrast to the United States [5]. The diagnosis of BC is considered a traumatic experience due to the problems associated with treatment, including physical problems such as fatigue, nausea and alopecia, and psychological problems such as depression, anxiety, and existential worry [6]. 30–50% of breast cancer survivors (BCSs) show symptoms of emotional distress such as depression and/or anxiety [3,7]. Depression and anxiety are classified in the American Psychiatric Association’s Diagnostic and Statistical Manual of Mental Disorders, Fifth Edition (DSM-5). The first symptom is characterized by a state of deep sadness and a loss of interest or pleasure that is present most of the day for a duration of least two weeks [8,9]. The second one has been defined as an emotional reaction characterized by feelings of tension, apprehension, nervousness and worry together with the activation of the sympathetic autonomic nervous system that has a functional value and biological utility [10].

During the first year, it is very common to develop both conditions, due to the severe emotional impact of the diagnosis and the chemotherapy (CH) and radiotherapy (RTH) treatment [11,12,13]. Its prevalence is 2–3 times higher than in the general population [11,12,13] and patients with BC have deeper depressive symptomatology compared to other types of cancer [11].

There are two surgery treatment options offered to patients. The first one is a mastectomy (MST), which is the most used treatment and where the entire breast is removed. It is the safest option because it increases the probability of survival up to 40%. The second one is breast-conserving therapy (BCT), in which only the tumor and the surrounding tissue are removed to prevent its growth. This option is the used when women want to preserve their body image. Women who had MST showed the greatest difficulties in understanding their own emotions [14].

Long-term survivors who have already completed the treatment phase can also suffer depression and anxiety [15]. Studies have even shown that survivors are unable to engage in stressful events 7 years after the end of the treatment [16]. In cancer care, there are five vital signs: temperature, respiration, heart rate, blood pressure and pain; together with this, emotional distress is added as a sixth vital sign [6]. Moreover, depression and anxiety are more common in patients with more pathologies or more advanced disease stages [11]. Depression and anxiety in cancer patients are caused by an inability to process the diagnosis and the experience of the disease [16], as well as a strong fear of death and concern for possible recurrence. Depression and anxiety are predictors of both [15]. In many cases, lack of support experienced by a patient leads to a vicious circle of negative emotions that makes the appearance of positive emotions impossible. As a consequence, there is an increased risk of developing psychopathologies that also produce physical symptoms of unknown origin [16].

The prevalence of depression and anxiety changes over the years due to the fluctuation in symptoms. In fact, there are studies that show rates of prevalence with wide range, with severe symptoms that can last for 15 years after the diagnosis. Concern about the risk of recurrence is the most repeated feeling among survivors with anxious-depressive symptoms [17,18].

It is very important to know which are the most recurrent factors associated with BC in patients suffering from depression or anxiety in order to establish more effective intervention programs for their psychological wellbeing. According to several studies [2], mortality can be reduced by 41% so it is very important for the healthcare system to identify early which patients develop depressive symptoms or which are at higher risk of developing such symptoms once cancer has been diagnosed [19]. Information can be provided to patients and their close relatives to mitigate their uncertainty and to enable informed decision making. There are a large number of variables that may explain the prevalence of depression and anxiety in BCSs, including demographic characteristics (age, gender, type of surgery, marital status, educational level and family history) [7,11,20,21]. Marital status has been the most studied factor due to the social support provided by the partner. In fact, depressive symptoms are developed by patients who have a weak social network [7], but the evidence on the factors associated with depression and anxiety in BCSs is unclear and in many cases contradictory.

The aim of this study was to identify factors related to depression and anxiety in mastectomized women BCSs. The information obtained in this study could expand knowledge of the characteristics of BCSs in order to adapt the treatment at a multidisciplinary level, according to the risk of suffering from depression or anxiety. In this way, the patient will be accompanied during the development of the disease by receiving the necessary psychological support or any other service that they require. This implementation will relieve or avoid these disorders.

## 2. Materials and Methods

### 2.1. Participants

A cross-sectional study was conducted in the state of Sonora, Mexico. The sample consisted of 198 women between the ages of 30 and 80 who had undergone surgery as part of BC treatment and live in the Mexican state. All of them participated in the study voluntarily and provided written informed consent after explanation of the nature and intention of the study.

### 2.2. Procedure

Participants were recruited according to the following inclusion criteria: be fluent in Spanish, be over 18 years of age at the time of filling out the questionnaire, had undergone radical mastectomy or breast-conserving surgery at least 9 months prior to completing the questionnaires, had a confirmed diagnosis of breast cancer and had been discharged by the surgeon. The main exclusion criteria were being pregnant at the time of evaluation and had undergone bilateral mastectomy as part of treatment.

The sample was divided into BCT and MST options, which are the two most common groups of surgery and encompass other types of subsequent surgeries [22], including breast reconstruction, which was grouped into the MST group because it is not covered by the Mexican healthcare system. For this reason, and due to the differences among them, it was decided to group them in this way. It was the healthcare professional who indicated to which group each woman belonged.

The study was approved by the Bioethics Committee on Human Research of the Department of Medicine and Health Sciences of the University of Sonora (DMCS/CBIDMCS/D-50). All of the women who participated in the study came from different hospitals in the province of Sonora that had medical speciality in oncology.

The evaluation of participants was performed on two occasions. Firstly, we administered a questionnaire of our own elaboration to obtain sociodemographic and personal data. In particular, the following data were collected: age classified by ranges (<50 years. between 50 and 65 years and >65 years), marital status (with or without a partner at the time of the evaluation), employment status at the time of the evaluation (employed, unemployed, disabled or retired), level of studies at the time of the evaluation (basic level between primary and secondary school, upper secondary level having completed high school, higher level having a university degree or a postgraduate degree and a level classified as “other” where technical studies or refresher courses were included), what type of surgery they have had (radical or conservative), whether they have had breast reconstruction (yes or no), family history of cancer (no history, first-degree history or second-degree history).

Secondly, a health professional collected the following data: type of surgery (radical or conservative mastectomy), the type of treatment they were receiving (RTH, CH, hormone therapy, targeted therapy or immunotherapy) and cancer stage (0, I, II, II, III, IV).

### 2.3. Assessments

The 14-item Hospital Anxiety and Depression Scale (HADS) [19] was used to measure the two dependent variables of the study, depression and anxiety. The scale is a 14-item self-report questionnaire which consist of seven items to assess anxiety (HADS-A subscale) and seven items to evaluate depression (HADS-D subscale) [20]. Each of the items are scored from 0 to 3, so the scores for both subscales range from 0 to 21. The questionnaire has shown high reliability for both subscales, with a mean Cronbach’s α of 0.83 [21]. The two subscales, anxiety and depression, are independent measures and divided into three stages: a score on either subscale between 0–7 is considered non-significant and classified as “normal”, between 8–10 is considered subclinically significant and classified as “symptomatic”, and between 11–21 is considered a clinically meaningful symptomatology of depression or anxiety and classified as “pathological”. The scale HADS has been administered in numerous studies in people with oncology treatment, demonstrating its validity and accuracy. It has been translated into Spanish [22], validated in the Mexican population [23] and is the gold standard for the assessment of these mental disorders [21,24].

Only one limitation on the scale was found. The questionnaire measures self-admitted anxiety and depression, so if the woman is in a phase of acceptance and she is actively fighting to overcame the disease, the scale will not assess the symptomatology correctly [15]. No participants were excluded as the entire sample (*n* = 198) replied to the questionnaire.

### 2.4. Statistical Analysis

Data are presented as number of cases (% of total) and as mean ± standard deviation (SD). Statistical analysis was performed with the software SPSS version 25 (IBM-Inc., Chicago, IL, USA). Statistical significance was determined with a *p*-value < 0.05.

Groupings were made according to data collected at the time of assessment. The variable age was divided into <50 years vs. ≥50 years because it is the median age of menopause in Mexico [25,26] and in the rest of the world [27] and also because BCSs under the age of 50 are the ones who suffer more depressive and anxiety symptoms [28,29]. The variable years since diagnostic or elapsed years was divided into <5 years vs. ≥5 years. Based on this criterion, doctors are able to estimate whether the cancer has been overcome and to aromatase inhibitor treatments and drugs such as tamoxifen, which are prescribed for a period of 5 years after the last CH or RTH [30]. The following variables were also classified: receiving treatment (receiving treatment vs. without receiving treatment), medical history (with medical history vs. without medical history), marital status (with partner vs. without partner), employment status (employee vs. unemployed), and educational level (up to high school vs. university or higher).

The differences in the total score of depression and anxiety between groups were tested by bivariate analysis, performing a t-test for independent samples, with the group to which each patient belonged as a fixed factor.

In addition, magnitude-based inference (MBI) was used for analysis to determine likelihood of the beneficial/trivial/harmful effect of the variables under study on the depression and anxiety subscales, using a dedicated spreadsheet following the terms and rules specified by Batterham and Hopkins [31,32,33]. Inferences were based on the confidence interval range of 90% to the smallest clinically meaningful effect to be positive, trivial or negative. Unclear results are reported if the observed confidence interval overlaps both positive and negative values. The levels of likely and compatibility are determined as follows: <0.5%, most unlikely; 0.5–5%, very unlikely; 5–25%, unlikely; 25–75%, possibly; 75–95%, likely; 95–99.5% very likely; >99.5%, most likely.

## 3. Results

Demographic and descriptive characteristics of the 198 women who participated in the study are shown in Table 1. The mean age was 52.9 years.

The assessment of depression and anxiety was carried out with the scale HADS, as detailed in Table 2. Data revealed that 187 women out of 198 (94.44%) of the total sample scored ≥8 points on the subscale HADS-A, which means that they had some anxiety symptoms. A total of 70.20% of these women scored between 11 and 21, which indicates pathological symptomatology (data no presented). The highest scores in HADS-A were in the variable without partner (12.54 ± 2.75) and in the group of ≥5 years after diagnosis of the disease (12.49 ± 2.78), and the lowest scores of anxiety were in the group of women without treatment (11.53 ± 2.86) and women <50 years (11.43 ± 2.99).

In the HADS-A subscale, a significant relationship was found in the variable years elapsed since diagnosis (*p* = 0.027) with a score of 11.54 ± 2.63 in <5 years and 12.49 ± 2.78 in ≥5 years, where MBI revealed 0.0/15.6/84.4% chance of beneficial/trivial/harmful effect, indicating that the passage of time is harmful for those suffering from anxiety. In the marital status variable (*p* = 0.031) the score of without partner was 12.54 ± 2.75 and the score for with partner was 11.57 ± 2.65. MBI developed 82.8/17.1/0.0% possibilities of beneficial/trivial/harmful effect, highlighting that having a partner is beneficial for mitigating anxiety.

MBI revealed 0.1/44.6/55.3% chances of beneficial/trivial/harmful effect of age on anxiety and 0.2/52.9/46.8% chances of beneficial/trivial/harmful effect of treatment on anxiety. On the other hand, MBI revealed 43.7/55.6/0.8% and 47.1/50.8/0.7% possibilities of beneficial/trivial/harmful effect of type of surgery and family history on anxiety, respectively.

One hundred thirty-seven women out of 198 (69.18%) of the total sample scored ≥ 8 points in the subscale HADS-D, which measures depression symptoms. Of these women, 10.60% were classified as pathological (data not presented). The highest scores of depression were found in women without medical history (8.61 ± 1.85) and in employed women (8.61 ± 1.68); the lowest scores were in unemployed participants (7.89 ± 2.11) and with medical histories (8.01 ± 2.04). In the HADS-D subscale, significant values were only found in the variable employment situation (*p* = 0.044) with a score of 7.89 ± 2.11 for unemployed and 8.61 ± 1.68 for employee. MBI developed 83.4/16.4/0.1% possibilities of beneficial/trivial/harmful effect, indicating that being unemployed is beneficial for not suffering from depression. MBI revealed 76.7/23.2/0.2% chances of beneficial/trivial/harmful effect of family history on depression and 26.4/69.3/4.3 possibilities of beneficial/trivial/harmful effect of marital status on depression.

## 4. Discussion

More than 94% of the BCSs who participated in our study showed symptoms of anxiety and more than 69% showed symptoms of depression, with 70% and 10% showing pathological symptoms, respectively. These data indicated a high prevalence of symptoms in these patients, where the side effects of the medication and the diagnosis of the disease were the main triggers. In this case, the identification and study of the demographic variables and characteristics of each patient are essential for better clinical approaches in the health system.

With regard to BCSs’ age variable, no significant results were found. For this reason, we performed a MBI analysis to define more objectively the possibilities of age affecting the development of depression and anxiety. Data revealed that older age (≥50 years) results in 55.3% chance of a worse anxiety score, with 44.6% and 0.1% chances of trivial and beneficial effects. These results are in agreement with other studies showing that having a cancer diagnosis at an early age predicts better prognosis and therefore a better QOL [34]; nevertheless, there are a large number of studies that concluded no difference between age and depressive or anxious symptoms [28].

Marital status showed that there was a direct correlation between being with or without a partner and the possibility of suffering from depression or anxiety in BCSs; Data showed significantly higher values in BCSs without a partner. Women with partners resulted in 26.4% and 82.2% chance of improvement on the anxiety and depression scales, respectively. These results are in agreement with previous studies which showed that the social role and socio-affective support provided by a partner is key to lower symptomatology [6,35], even indicating that a BCS without a partner at the time of diagnosis could be 20% more likely to suffer from depression or anxiety [36].

There are other studies that do not focus on marital status but on the social support that the BCSs receive, specifically from partners and family, as the most important. When BCSs have one or more children, studies show that it can be a predictor of deeper depression [37,38,39,40,41].

The type of surgery performed as part of the treatment does not indicate significant results, although the MBI analysis shows that women undergoing conservative surgery have 43.7% beneficial chances on the anxiety scale, and 55.6% trivial effect. These results are consistent with other studies indicating that women undergoing MST reported worse body image, greater pain, higher depression, higher anxiety and hopelessness scores than women undergoing BCT [14,15].

Regarding the variable receiving treatment, our results show no significance, but inference reveals 46% of harmful possibilities on the anxiety scale and 24.8% on the depression scale, indicating that BCSs who received treatment (CH or RTH) show higher levels of depression and anxiety. The results are in agreement with a large number of studies which state that the side effects of treatment, such as nausea, vomiting, fatigue and deterioration of body image caused by alopecia, skin or nail problems, are one of the triggers for this symptomatology [12,34,37,38]. During the course of the treatment, 21–54% suffered from anxiety symptoms and 12–31% suffered from depressive symptoms [39]. This symptomatology does not disappear at the end of the treatment, and there may be higher levels of depression and anxiety 5 years after having finished the treatment [16,38,40]. Due to large scientific evidence, it is important to emphasize that BCSs who have received any type of treatment should receive more attention from the health system [40].

Our study does not show significant findings on the educational level variable and the analysis of inference shows unclear results. The majority of the literature showed that a higher level of education was associated with higher levels of depression and anxiety at the start of treatment. This level decreased over time, which may be due to the fact that the adaptation process in their case is more extensive [39,41,42].

The time elapsed since the diagnosis to the assessment of depression and anxiety is a very important study variable to know whether scores on both scales changed over time. Our study shows that the more time has elapsed, the higher the significant score on the HADS-A subscale. These results are in agreement with other studies which have indicated levels increasing over time [16]. In a recent German study, results revealed that BCSs show significantly higher HADS scores 5–6 years after diagnosis than after 40 weeks [15]. The reason can be found in the time of diagnosis. After diagnosis, anxiety levels increase; however, when a patient has adapted, the level of anxiety decreases because at this stage the disease is usually defeated and treatment is complete so positive emotions such as relief predominate. Nevertheless, after a long period of time, symptoms can become more severe as BCSs deal with new emotions and with social, economic and physical concerns, which leads to an intensification of depressive and anxious levels [15,43,44].

Family history of cancer is a major risk factor for developing the disease, but it has not to be a factor for depression or anxiety. There are very few studies that have studied this variable. An article carried out with Latina BCSs showed no significant data between family history and symptoms of depression [45]. In contrast, in a study conducted with Turkish BCSs, data showed significant results between having depression and family history [46]. Our results do not show significant data. The MBI analysis indicates a 76% of beneficial chance, with 23.2% of trivial effect on the depression scale and a 47% of beneficial chance, with 50.8% of trivial effect on the anxiety scale.

Finally, the variable of having a job during the diagnosis has not been sufficiently studied. Our findings indicate significantly higher symptomatologic values in BCSs who worked, which means it an important risk factor for depression. These results are in agreement with a study which concluded that BCSs who were employed at the time of diagnosis and continued to work had higher levels of anxiety [42].

The reason for this result is likely because BCSs are worried about returning to work. Although there are not many studies that show significant data, there are indications that a large number of BCSs are not able to work as they are too ill [39]. At the same time, BCSs are overwhelmed when they are the economic breadwinner of the family.

It should be noted the variables of employment status and years elapsed since diagnosis as two predictors for depression and anxiety. Vahdanina et al. [47] showed that depression and anxiety decreased over time.

If we compare our results obtained in the HADS with other BCS populations who have also used this assessment, we can see that our score levels are higher than Southeast Asian populations, where the prevalence of anxiety ranges from 7% to 88% and depression levels ranging from 3% to 65.5%. Turkish and Iranian BCSs showed the prevalence of suffering depression and anxiety was below 50% [44]. In other cancers such as prostate cancer, the existence of depressive or anxiety symptoms among survivors is 12.5% and 23.3%, respectively [44]; in the UK, anxiety and depressive symptoms in the survivor population of any type of cancer does not exceed 30% [48].

It has to be emphasized that the prevalence of depression and anxiety in a general population compared with BCSs is much lower. Approximately 10% of the German population suffer from these symptoms; in countries such as USA or Greece, it is close to 12% [49]. If we compare the data with other pathologies such as Type 2 Diabetes, it is observed that the percentages of depression and anxiety in Mexico range from 28% to 55% [50] and reach around 70% in the population with peripheral neuropathy in Tunisia [51]. These data show the high prevalence of depression and anxiety in our sample compared to other populations. This may be because women with BC go through severe treatments with a large number of side effects and it is the only cancer in which amputation is performed, restricting daily activities [12,43].

Finally, regarding the limitations of this study, it was not possible to compare the results of studies that include women who had received psychological treatment, because they were not available. These results cannot be generalized to the entire population of Mexico or to the world population, as the sample was only taken from one state in Mexico. It may be important for future research to take into account the impact that culture may have on these variables. In addition, the use of self-administered questionnaires may result in a bias when interpreting the results, so they should be taken with caution, despite being a questionnaire with good properties. The lack of randomization and the absence of a response rate should also be considered. Further research is needed on this issue as it affects so many women around the world.

## 5. Conclusions

Our study has shown that BCSs under 50 years of age who were receiving some form of treatment, with no family history, no partner, were employed, with more than a high school education and with more than 5 years since diagnosis could have higher rates of clinical depression. On the other hand, a BCS over 50 years old who was receiving some form of treatment, with no family history, no partner, has a job, with more than a high school education and with more than 5 years since diagnosis, could have higher rates of clinical anxiety, although the factors associated with depression and anxiety vary depending on the primary endpoints of the studies. Likewise, a small relationship has been observed between depression and employment situation, and between anxiety with elapsed years and marital status. Further studies should be undertaken to categorize the profiles of women who are more likely to have deeper depressive or anxious symptoms after diagnosis, during treatment or after the end of treatment. It is necessary to identify the factors associated with depression and anxiety in these women to establish a personalized and individualized action protocol so these patients receive the appropriate care when necessary. It would improve their healthcare and therefore increase their chances of survival. These protocols could include the prescription of personalized physical exercise, therapies with psycho-oncology professionals, self-care activities and talks with women BCSs to advise them and share their experiences [52,53].

Some studies showed that BCSs who receive psychotherapy reduce the risk of recurrence by 43% [2]. This is of vital importance to detect which cancer patient profile may be more likely to develop depressive or anxious symptoms during diagnosis, subsequent treatment and recovery, and also to improve the QOL of BCSs once they have overcome cancer.

Further studies are needed to analyze which variables may be predictors of depression and anxiety in BCSs, with larger sample sizes and with greater number of variables analyzed. This will facilitate the categorization of cancer profiles, the implementation of personalized protocols for each patient, and the offer of psychotherapy or other services required by the patient. As a consequence, the risk of mortality will be reduced.

## Figures and Tables

**Table 1 ijerph-20-02881-t001:** Clinical and demographic characteristics of the study group.

Variables		Total (n = 198)
Age		52.9 ± 8.6
Age	<50 years	86 (56.56%)
	≥50years	112 (43.43%)
Elapsed years	<5years	145 (73.23%)
	≥5years	53 (26.76%)
Receiving Treatment	Without receiving treatment	103 (52.02%)
	Receiving treatment	95 (47.97%)
Type of surgery	Breast-conserving therapy	126 (63.63%)
	Mastectomy	72 (36.36%)
Family history	Without family history	49 (24.74%)
	With family history	149 (75.25%)
Marital status	Without partner	46 (23.23%)
	With partner	152 (76.76%)
Employment situation	Employee	57 (28.78%)
	Unemployed	141 (71.21%)
Level of education	Up to high school	117 (59.09%)
	University or higher	81 (40.90%)

**Table 2 ijerph-20-02881-t002:** Association between characteristics, depression and anxiety in breast cancer patients (HADS).

Total Score	Variable	Group		*p*	% Difference (90% CI)	Qualitative Assessment
	Age	<50 years (*n* = 86)	≥50 years (*n* = 112)			
Depression		8.17 ± 2.18	8.15 ± 1.88	0.938	0.02 (−0.48 to 0.50)	Unlikely
Anxiety		11.43 ± 2.99	12.07± 2.42	0.097	−0.64 (−1.30 to 0.00)	Most unlikely
	Elapsed years	<5 years (*n* = 145)	≥5 years (*n* = 53)			
Depression		8.14 ± 1.94	8.23 ± 2.22	0.785	0.08 (−0.62 to 0.45)	Unlikely
Anxiety		11.54 ± 2.63	12.49 ± 2.78	0.027	−0.95 (−1.70 to −0.25)	Most unlikely
	Receiving treatment	Without receiving treatment (*n* = 103)	Receiving treatment (*n* = 95)			
Depression		8.06 ± 2.07	8.27 ± 1.94	0.452	0.22 (−0.69 to 0.26)	Very unlikely
Anxiety		11.53 ± 2.86	12.07 ± 2.50	0.160	−0.54 (−1.20 to 0.09)	Most unlikely
	Type of surgery	Mastectomy (*n* = 72)	Breast-conserving therapy (*n* = 126)			
Depression		8.18 ± 1.94	8.15 ± 2.05	0.920	0.03 (−0.46 to 0.52)	Unlikely
Anxiety		12.08 ± 2.64	11.62 ± 2.72	0.253	0.46 (−0.20 to 1.10)	Possibly
	Family history	Without medical history (*n* = 49)	With medical history (*n* = 149)			
Depression		8.61 ± 1.85	8.01 ± 2.04	0.070	0.60 (0.10 to 1.10)	Likely
Anxiety		12.12 ± 2.39	11.68 ± 2.79	0.325	0.44 (−0.30 to 1.20)	Possibly
	Marital status	Without partner (*n* = 46)	With partner (*n* = 152)			
Depression		8.30 ± 2.03	8.12 ± 2.01	0.584	0.19 (−0.37 to 0.75)	Possibly
Anxiety		12.54 ± 2.75	11.57 ± 2.65	0.031	0.98 (0.20 to 1.70)	Likely
	Employment situation	Employee (*n* = 57)	Unemployed (*n* = 141)			
Depression		8.61 ± 1.68	7.89 ± 2.11	0.044	0.64 (0.12 to 1.20)	Likely
Anxiety		11.81 ± 2.46	11.79 ± 2.79	0.963	0.02 (−0.70 to 0.70)	Unlikely
	Level of education	Up to secondary school (*n* = 117)	From secondary school (*n* = 81)			
Depression		8.16 ± 2.11	8.16 ± 1.86	0.995	0.01 (−0.50 to 0.50)	Unlikely
Anxiety		11.75 ± 2.82	11.85 ± 2.53	0.799	−0.10 (−0.70 to 0.50)	Very unlikely

Data are presented as mean ± standard deviation. CI= Confidence interval.

## Data Availability

Not applicable.

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
