# Peer review of "Related Factors with Depression and Anxiety in Mastectomized Women Breast Cancer Survivors"

_ijerph, 2023, doi:10.3390/ijerph20042881_

Round 1

Reviewer 1 Report (Previous Reviewer 3)

---------------------------------------------------------------------

Manuscript ID: ijerph-2160328

Related factors with depression and anxiety in mastectomized women breast cancer survivors

---------------------------------------------------------------------

The authors have modified the manuscript “Related factors with depression and anxiety in mastectomized women breast cancer survivors”. This new version submitted unfortunately has most of the flaws of its previous versions. I must insist on them before considering the manuscript for publication.

Effect sizes: In my very first review, I noted that Ferguson (2009) did not propose thresholds for small, medium, and large effect sizes. He merely summarized values from other sources. Therefore, there is no such thing as "Ferguson’s criteria". The mention to Ferguson's criteria disappeared in your second version, but you have reinstated it. What he actually did is to insist on the importance of context in interpreting effect size indices. That is what I meant, in my first review, when I said “interpret them in a meaningful way”. Arbitrary labels should not be interpreted as universal cut-off points. The practicality to which Ferguson refers is context-driven. Small values can convey the idea of relevant effects, and vice versa.

I may sound harsh with all this but, actually, I am giving you the opportunity to make sense of your results. If you go by your own words, you have found effect size indices below practicality. That is, the statistical significance of those tests may be due to sample size, low standard error of the test estimates, or the effect of some confounding variables. You state, on p. 5, that "we should interpret these results with caution". In fact, you should interpret these results in a completely different way. You have not found any relationship of practical relevance. Or have you? This is why an adequate contextual interpretation of effect sizes is crucial to your study.

Other references on this subject may be consulted if necessary (see, for instance, Cortina & Landis, 2009, Cumming, 2013, Sánchez-Iglesias et al., 2023).

Cortina, J.M.; Landis, R.S. (2009). When small effect sizes tell a big story, and when large effect sizes don’t. In C. E. Lance, & R. J. Vandenberg (Eds.). Statistical and methodological myths and urban legends: Doctrine, verity and fable in the organizational and social sciences. Taylor & Francis.

Cumming, G. (2013). The New Statistics. Psychological Science, 25(1), 7–29. https://doi.org/10.1177/0956797613504966

Sánchez-Iglesias, I., Saiz, J., Molina, A. J., & Goldsby (2023). Reporting and Interpreting Effect Sizes in Applied Health-Related Settings: The Case of Spirituality and Substance Abuse. Healthcare, 11(1), 133. https://doi.org/10.3390/healthcare11010133

In addition, my question about how you computed η2 (and, for that matter, observed power) for a t-test using SPSS remains unanswered. SPSS does not report such statistics.

Assumptions of the statistical tests: I must insist on this for the third time. Your most recent answer in this regard is that you added new information, but the sentence you refer to was already in previous versions, and it –still– does not address the question.

Please, consult a methodologist to learn about statistical assumptions for statistical tests, so you can check and report them in the manuscript for the t-tests and regression models. Also, as I stated in my previous review, much more detail is needed about the regression model. Why are the p-values of the significant independent values identical to those of the t-test? Is it because you have performed several simple regression analyses, one for each independent variable? What do simple regression models offer beyond t-tests?

Please check these comments carefully (and any others not yet answered from my previous reviews), and consult a methodology specialist to resolve these issues.

Author Response

Mrs. Jessica Fernández Solana

Department of health sciences

University of Burgos, Paseo Comendadores s/n.

Burgos, 09001, Spain

Tel. (+34) 947499108

27-01-2023

IJERPH.  Subject: Submissions Needing Revision

Dear editor.

Thank you very much for inviting us to submit our response to reviewers for our manuscript (ijerph-2160328) entitled: “Related factors with depression and anxiety in mastectomized women breast cancer survivors”

We have checked our manuscript according to the Academic Editor, the reviewers’ comments and the Journal requirements. We have also responded to some comments from reviewers point by point).

We would be very grateful if you could consider our manuscript to be published in your journal.

Yours sincerely,

Jessica Fernández Solana, OT, PT

  1. Response to Reviewer 1:

First of all, we would like to express our sincere gratitude for all comments and suggestions received from the Reviewer 1. This information has certainly enriched the text for its best understanding, thank you very much indeed. We have clarified the reviewer1’s questions. We have introduced the required changes both in our answers to the specific comments and in the final manuscript V2.

The authors have modified the manuscript “Related factors with depression and anxiety in mastectomized women breast cancer survivors”. This new version submitted unfortunately has most of the flaws of its previous versions. I must insist on them before considering the manuscript for publication.

Effect sizes: In my very first review, I noted that Ferguson (2009) did not propose thresholds for small, medium, and large effect sizes. He merely summarized values from other sources. Therefore, there is no such thing as "Ferguson’s criteria". The mention to Ferguson's criteria disappeared in your second version, but you have reinstated it. What he actually did is to insist on the importance of context in interpreting effect size indices. That is what I meant, in my first review, when I said “interpret them in a meaningful way”. Arbitrary labels should not be interpreted as universal cut-off points. The practicality to which Ferguson refers is context-driven. Small values can convey the idea of relevant effects, and vice versa.

I may sound harsh with all this but, actually, I am giving you the opportunity to make sense of your results. If you go by your own words, you have found effect size indices below practicality. That is, the statistical significance of those tests may be due to sample size, low standard error of the test estimates, or the effect of some confounding variables. You state, on p. 5, that "we should interpret these results with caution". In fact, you should interpret these results in a completely different way. You have not found any relationship of practical relevance. Or have you? This is why an adequate contextual interpretation of effect sizes is crucial to your study.

Response: Thank you for your comments. We have amended this issue (See lines 190-198, 223-250 and Table 2)

In addition, my question about how you computed η2 (and, for that matter, observed power) for a t-test using SPSS remains unanswered. SPSS does not report such statistics.

Response: Thank you for your comment. We have made changes in the manuscript.

Assumptions of the statistical tests: I must insist on this for the third time. Your most recent answer in this regard is that you added new information, but the sentence you refer to was already in previous versions, and it –still– does not address the question.

Response: Thank you for your comment. We have made changes in the manuscript.

Please, consult a methodologist to learn about statistical assumptions for statistical tests, so you can check and report them in the manuscript for the t-tests and regression models. Also, as I stated in my previous review, much more detail is needed about the regression model. Why are the p-values of the significant independent values identical to those of the t-test? Is it because you have performed several simple regression analyses, one for each independent variable? What do simple regression models offer beyond t-tests?

Response: Thank you for your comment. We have taken the request to a methodology.

We hope we have now answered all your comments and we are looking forward to hearing from you again.

Jessica Fernández Solana, OT, PT

Reviewer 2 Report (New Reviewer)

Thank you for this interesting manuscript. It definitely points out areas that need to be improved in the treatment of breast cancer patients. The following are my questions, comments, and/or suggestions:

1. There needs to be significant editing done to the manuscript. I am not going to go through and point them all out, but there are many run-on sentences which make it difficult to read and understand the manuscript.

2. There are also quite a few type-o's that need to be corrected, words that are left out, misplaced commas, incomplete sentences, etc.

3. In addition to the run-ons, there are also several times that paragraphs were made up of only one sentence. The standard is usually to have at least three sentences per paragraph.

4. Please be consistent with tenses. For example, once you start talking about what happened/what you found, the document should be written in the past tense.

5. Page 1, line 45 - please capitalize and include the full name of the DSM.

6. Page 2, line 53 - you mentioned patients undergoing chemotherapy. What about those undergoing radiation treatments?

7. Page 2, lines 68-71, sentence beginning "Depression and anxiety in cancer patients..." - this sentence is a bit confusing. Perhaps it could be reworded.

8. Pages 2-3, lines 98-100 - this sentence does not appear to be a full sentence.

9. Page 3 - there is some inconsistency. In line 104 it was stated that the participants had all undergone mastectomies; however, in line 111 it was stated that they had either undergone radical mastectomies or breast conserving surgery. Can you please clarify? Also, can you please clarify whether or not your patients had a single mastectomy or a bilateral mastectomy (I apologize if this was there and I missed it).

10. Page 3, lines 115-118 - this sentence is somewhat confusing.

11. Page 3m, lines 137-138 - I'm curious. Why did you ask patients about their perceived sense of humor and how did this fit into what you were studying?

12. Page 4, line 158 - can you please explain what is meant by the woman being "in defence of the disease"?

13. Pages 4-5, Results - you may want to consider including all of the information about HADS-A together and all of the information about HADS-D together; or group the information on each topic (i.e. family history) together in one paragraph. It might help this section flow better.  

14. Page 6, lines 238-239 - did you mean that younger survivors tend to have higher levels of depression and lower anxiety scores than older women? This sentence was just a bit confusing.

15. Page 7, lines 287-292 - this is a long sentence that is confusing. 

16. Page 8, lines 327-238 - this is a confusing sentence. I'm also not sure what is meant by "any type of symptom".

17. Please review your references for accuracy and consistency. For example, in some, every word in the title of the article is capitalized and in others only the first word is capitalized. Also, in some the full name of the journal is written out and in others it is abbreviated.

P

Author Response

Mrs. Jessica Fernández Solana

Department of health sciences

University of Burgos, Paseo Comendadores s/n.

Burgos, 09001, Spain

Tel. (+34) 947499108

27-01-2023

IJERPH.  Subject: Submissions Needing Revision

Dear editor.

Thank you very much for inviting us to submit our response to reviewers for our manuscript (ijerph-2160328) entitled: “Related factors with depression and anxiety in mastectomized women breast cancer survivors”

We have checked our manuscript according to the Academic Editor, the reviewers’ comments and the Journal requirements. We have also responded to some comments from reviewers point by point).

We would be very grateful if you could consider our manuscript to be published in your journal.

Yours sincerely,

Jessica Fernández Solana, OT, PT

  1. Response to Reviewer 2:

First of all, we would like to express our sincere gratitude for all comments and suggestions received from the Reviewer 2. This information has certainly enriched the text for its best understanding, thank you very much indeed. We have clarified the reviewer2’s questions. We have introduced the required changes both in our answers to the specific comments and in the final manuscript V2.

1.There needs to be significant editing done to the manuscript. I am not going to go through and point them all out, but there are many run-on sentences which make it difficult to read and understand the manuscript.

Response: Thank you very much for pointing this out, we have made an exhaustive revision of the manuscript with the help of a professional.

  1. There are also quite a few type-o's that need to be corrected, words that are left out, misplaced commas, incomplete sentences, etc.

Response: Thank you very much for pointing this out, we have revised the manuscript and changed the way it is written.

3.In addition to the run-ons, there are also several times that paragraphs were made up of only one sentence. The standard is usually to have at least three sentences per paragraph.

Response: Thank you very much for pointing this out, we have modified this section making it easier to read it.

  1. Please be consistent with tenses. For example, once you start talking about what happened/what you found, the document should be written in the past tense.

Response: Thank you very much for pointing this out, we have revised the manuscript and changed the way it is written.

  1. Page 1, line 45 - please capitalize and include the full name of the DSM.

Response: Thank you very much for pointing this out, we have added this information in the manuscript (See lines 46-47) “American Psychiatric Association's Diagnostic and Statistical Manual of Mental Disorders, Fifth Edition (DSM-5)”.

  1. Page 2, line 53 - you mentioned patients undergoing chemotherapy. What about those undergoing radiation treatments?

Response: Thank you very much for pointing this out, we have added this information in the manuscript (See line 53) “and radiotherapy (RTH) treatment.”.

  1. Page 2, lines 68-71, sentence beginning "Depression and anxiety in cancer patients..." - this sentence is a bit confusing. Perhaps it could be reworded.

Response: Thank you very much for pointing this out, we have modified the section making it easier to read (See lines 66-72) “In cancer care, there are five vital signs which are: temperature, respiration, heart rate, blood pressure and pain; together with this, emotional distress is added as a sixth vital sign. Moreover, depression and anxiety are more common in patients with more pathologies or more advanced disease stage. Depression and anxiety in cancer patients are caused by an inability to process the diagnosis and the experience of the disease together with a strong fear of death and a concern of a possible recurrence”.

  1. Pages 2-3, lines 98-100 - this sentence does not appear to be a full sentence.

Response: Thank you very much for pointing this out, we have modified the section making it easier to read (See lines 97-103) “The information obtained in this study could expand knowledge of the characteristics of BCS in order to adapt the treatment, at a multidisciplinary level, according to the risk of suffering depression or anxiety. In this way, the patient will be accompanied during the development of the disease by providing all necessary psychological support or any other service that the patient requires. This implementation will relieve or avoid these disorders”.

  1. Page 3 - there is some inconsistency. In line 104 it was stated that the participants had all undergone mastectomies; however, in line 111 it was stated that they had either undergone radical mastectomies or breast conserving surgery. Can you please clarify? Also, can you please clarify whether or not your patients had a single mastectomy or a bilateral mastectomy (I apologize if this was there and I missed it).

Response: Thank you very much for pointing this out, we have modified this information in the manuscript, and added a new point in the exclusion criteria (See lines 108,116-118) “The main exclusion criteria were being pregnant at the time of evaluation and having undergone bilateral mastectomy as part of treatment”.

  1. Page 3, lines 115-118 - this sentence is somewhat confusing.

Response: Thank you very much for pointing this out, we have modified the section making it easier to read (See lines 113-118) “Participants were recruited according to the following inclusion criteria: be fluent in Spanish, over 18 years of age at the time of filling out the questionnaire, having undergone radical mastectomy or breast-conserving surgery at least 9 months prior to completing the questionnaires, confirmed diagnosis of breast cancer and having been discharged by the surgeon. The main exclusion criteria were being pregnant at the time of evaluation and having undergone bilateral mastectomy as part of treatment”.

  1. Page 3m, lines 137-138 - I'm curious. Why did you ask patients about their perceived sense of humor and how did this fit into what you were studying?

Response: Thank you very much for pointing this out, we have removed this section (See line 140). These data are part of another research that will be carried out in the future by the same research group and that were taken at the same time as the research that makes up this manuscript.  We are sorry for having introduced the description of these variables without having used them in the manuscript.

  1. Page 4, line 158 - can you please explain what is meant by the woman being "in defence of the disease"?

Response: Thank you very much for pointing this out, we have added this information in the manuscript (See lines 165,169) “Only a limitation on the scale was found. The questionnaire measures self-admitted anxiety and depression, so if the woman is in a phase of acceptance and she is actively fighting to overcame the disease, the scale will not assess the symptomatology correctly [15]. No participants were excluded as the entire sample (n=198) replied to the ques-tionnaire.”.

  1. Pages 4-5, Results - you may want to consider including all of the information about HADS-A together and all of the information about HADS-D together; or group the information on each topic (i.e. family history) together in one paragraph. It might help this section flow better.  

Response: Thank you very much for pointing this out, we have modified the section making it easier to read (See lines 219-250).

  1. Page 6, lines 238-239 - did you mean that younger survivors tend to have higher levels of depression and lower anxiety scores than older women? This sentence was just a bit confusing.

Response: Thank you very much for pointing this out, we have modified the section making it easier to read (See line 291) “With regard to the BCS age variable, no significant results were found”.

  1. Page 7, lines 287-292 - this is a long sentence that is confusing. 

Response: Thank you very much for pointing this out, we have modified the section making it easier to read (See lines 334-340) “During the course of the treatment, the 21-54% suffered anxiety symptoms and the 12-31% depressive symptoms [39]. This symptomatology does not disappear at the end of the treatment and there may be higher levels of depression and anxiety 5 years after have finished the treatment [16, 38, 40]. Due to the large scientific evidence, it is important to emphasise that those BCS who have received any type of treatment should receive more attention from the health system [40]”.

  1. Page 8, lines 327-238 - this is a confusing sentence. I'm also not sure what is meant by "any type of symptom".

Response: Thank you very much for pointing this out, we have modified the section making it easier to read (See lines 410-411) “Approximately the 10% of the German population suffer from these symptoms and in countries such as USA or Greece close to 12% [49]”.

  1. Please review your references for accuracy and consistency. For example, in some, every word in the title of the article is capitalized and in others only the first word is capitalized. Also, in some the full name of the journal is written out and in others it is abbreviated.

Response: Thank you very much for pointing this out, we have made an exhaustive revision of the manuscript with the help of a professional.

We hope we have now answered all your comments and we are looking forward to hearing from you again.

Jessica Fernández Solana, OT, PT

Round 2

Reviewer 1 Report (Previous Reviewer 3)

---------------------------------------------------------------------

Manuscript ID: ijerph-2160328

Related factors with depression and anxiety in mastectomized women breast cancer survivors

---------------------------------------------------------------------

The authors have modified the manuscript “Related factors with depression and anxiety in mastectomized women breast cancer survivors”. Unfortunately, you have not addressed any of my earlier comments and suggestions. This version has even regressed to the flaws and shortcomings I pointed out in previous manuscripts.

You have chosen to ignore all my previous comments on the need to calculate and interpret effect sizes, and the convenience of adjusting a multiple linear regression model to establish patient profiles. You have also failed to address the assumptions of your statistical analyses. Instead of all of that, you have added a procedure called Magnitude-based inference (MBI). However, this method does not replace null hypothesis significance testing, effect size estimations and their interpretations in an adequate and meaningful context. In fact, one the references you have provided to justify the use of MBI states "We conclude that MBI has promoted small studies, promulgated a “black box” approach to statistics, and led to numerous papers where the conclusions are not supported by the data". Also “We found that MBI has done direct harm to the sports science and medicine literature by causing authors to draw overly optimistic conclusions from their data”. They even conclude that “Sports scientists should stop using MBI”. Have you knowingly chosen a faulty method, or failed to read your own references?

I have tried to be helpful to the authors in four different revisions of this manuscript, with indications that I thought were clear. At this point, all I hope is that you will review the entire study, with the help of an expert in research methods and statistics, before presenting these results again.

Reviewer 2 Report (New Reviewer)

Thank you for making the suggested revisions. It is a much stronger manuscript now. Congratulations!

This manuscript is a resubmission of an earlier submission. The following is a list of the peer review reports and author responses from that submission.

Round 1

Reviewer 1 Report

Thank you for offering me the opportunity to review the article entitled “Factors associated with depression and anxiety in mastectomised women breast cancer survivors”. This manuscript may be of interest because of the great psychological impact that patients suffer from the time they are diagnosed with breast cancer until survival. Despite the effort of the authors, I note many limitations in considering publication of this manuscript. For example, the information in the abstract differs from that provided by the methodology itself and the title of the present study.

Some errors in grammar are also observed. It is suggested to the authors that they can revise the syntactic and grammatical errors in the text, in addition to revising the formatting.

I suggest a few comments to improve this manuscript:

*Abstract

- The phrase “The data show significant results of some type of anxiety symptoms in more than of 94% of the participants, as well as depression symptoms in more than 69% of the participants, being considered pathological” is confusing. One of the reasons for not understanding the explanation of the results is the lack of numerical data on the statistical analysis performed.

- The analyses are performed based on the age of the patients and at no time is this stated in the abstract.

- It is suggested to provide conclusions that are valuable in clinical practice with women diagnosed with breast cancer or survivors according to the methodology proposed.

*Introduction

The idea is not understood in lines 40-41 page 1.  What do the authors mean by the sixth vital sign of cancer?

The important point of the introduction is the relationship between anxious and depressive symptomatology with medical variables such as the type of surgery performed. And, specifically, the relationship of adverse emotional symptomatology with mastectomy. There is a lot of literature on this subject to frame the research question in this section.

On the contrary, unnecessary information appears regarding infertility, lack of social support, quality of life... the authors give the reader to understand that they are going to establish a methodology in which they are going to relate these variables. However, this is not the case and the comprehension of the text is complicated.

I recommend the authors to rewrite the Introduction taking into consideration the research question: relationship between anxiety, depression and mastectomy. There is a lot of literature that relates the adverse emotional state to the type of surgery that has been performed and that has to do with body image distortion or alexithymia processes.

I suggest this study to review and remodel the Introduction:

Gutiérrez Hermoso, L., Velasco Furlong, L., Sánchez-Román, S., & Salas Costumero, L. (2020). The Importance of Alexithymia in Post-surgery. Differences on Body Image and Psychological Adjustment in Breast Cancer Patients. Frontiers in psychology11, 604004. https://doi.org/10.3389/fpsyg.2020.604004

*Method

Mastectomies can be classified into several types. What specific procedure was performed on the patients in the study? I suggest explaining this in the demographic and clinical results of the sample.

It is not clear whether the reliability index of the instruments is the one drawn from this sample or whether it is the one drawn from the validation of the questionnaire.

I do not understand the relationship between the Introduction and the Method. According to the statistical pathway, several groups are made according to age and other sociodemographic variables. However, this information does not appear in the previous theoretical framework. There is no connection of ideas.

*Results

They are well written and understandable. However, as I have previously indicated, they do not follow a sense in relation to the research question formulated nor to the title of this manuscript.

*Discussion

Some methodological limitations are lacking in relation to the type of study. For example, a cross-sectional study does not allow confirmation of hypotheses or maintenance of results over time.

The Discussion section could benefit from a paragraph of clinical implications beyond a sentence indicating the allocation of resources in the Health System. Specifically, what resources do you want to make available to patients with this problem?

Author Response

Mrs. Jessica Fernández Solana

Department of health sciences

University of Burgos, Paseo Comendadores s/n.

Burgos, 09001, Spain

Tel. (+34) 947499108

08-12-2022

IJERPH.  Subject: Submissions Needing Revision

Dear editor.

Thank you very much for inviting us to submit our response to reviewers for our manuscript (ijerph-2014157) entitled: “Quality of life in breast cancer survivors in relation to age, type of surgery and length of time since first treatment”

We have checked our manuscript according to the Academic Editor, the reviewers’ comments and the Journal requirements. We have also responded to some comments from reviewers point by point).

We would be very grateful if you could consider our manuscript to be published in your journal.

Yours sincerely,

Jessica Fernández Solana, OT, PT

  1. Response to Reviewer 1:

First of all, we would like to express our sincere gratitude for all comments and suggestions received from the Reviewer 1. This information has certainly enriched the text for its best understanding, thank you very much indeed. We have clarified the reviewer1’s questions. We have introduced the required changes both in our answers to the specific comments and in the final manuscript V2.

Thank you for offering me the opportunity to review the article entitled “Factors associated with depression and anxiety in mastectomised women breast cancer survivors”. This manuscript may be of interest because of the great psychological impact that patients suffer from the time they are diagnosed with breast cancer until survival. Despite the effort of the authors, I note many limitations in considering publication of this manuscript. For example, the information in the abstract differs from that provided by the methodology itself and the title of the present study.

Some errors in grammar are also observed. It is suggested to the authors that they can revise the syntactic and grammatical errors in the text, in addition to revising the formatting.

I suggest a few comments to improve this manuscript:

*Abstract

- The phrase “The data show significant results of some type of anxiety symptoms in more than of 94% of the participants, as well as depression symptoms in more than 69% of the participants, being considered pathological” is confusing. One of the reasons for not understanding the explanation of the results is the lack of numerical data on the statistical analysis performed.

- The analyses are performed based on the age of the patients and at no time is this stated in the abstract.

- It is suggested to provide conclusions that are valuable in clinical practice with women diagnosed with breast cancer or survivors according to the methodology proposed.

 Response: Thank you for your comment about this section. Changes have been made in relation to your comments.

“The results show that 94.44% and 69.18% of the women scored more than 8 points on HADS for anxiety and depression respectively; 70.20% and 10.60% were identified as pathological. The following variables were analyzed: age, time elapsed since the start of treatment, treatment, type of surgery, family history, marital status and employment status, of which the time elapsed since surgery, having a partner and employment show significant results being factors associated to depression and anxiety in this type of patients.”.

“In conclusion, it has been shown that BCS under 50 years of age, who have received some kind of treatment, without family history, without a partner, with a job, with more than secondary edu-cation and with more than 5 years since diagnosis, indicate higher rates of clinical depression. On the other hand, a BCS older than 50 years, who have received some kind of treatment, without family history, without a partner, with a job, with more than secondary education and with more than 5 years since diagnosis, indicate higher rates of clinical anxiety. The variables studied provide valuable information for healthcare systems to monitor more closely and reduce the risk of de-pression and/or anxiety in women with BC undergoing mastectomy”.

*Introduction

The idea is not understood in lines 40-41 page 1.  What do the authors mean by the sixth vital sign of cancer?

Response: Thank you for pointing this out. We have modified this section making it easier to read it. (See lines 60-66).

“There are five vital signs in cancer care which are temperature, respiration, heart rate, blood pressure and pain; although a sixth vital sign is added which is known as emotional distress”

The important point of the introduction is the relationship between anxious and depressive symptomatology with medical variables such as the type of surgery performed. And, specifically, the relationship of adverse emotional symptomatology with mastectomy. There is a lot of literature on this subject to frame the research question in this section.

On the contrary, unnecessary information appears regarding infertility, lack of social support, quality of life... the authors give the reader to understand that they are going to establish a methodology in which they are going to relate these variables. However, this is not the case and the comprehension of the text is complicated.

I recommend the authors to rewrite the Introduction taking into consideration the research question: relationship between anxiety, depression and mastectomy. There is a lot of literature that relates the adverse emotional state to the type of surgery that has been performed and that has to do with body image distortion or alexithymia processes.

I suggest this study to review and remodel the Introduction:

Gutiérrez Hermoso, L., Velasco Furlong, L., Sánchez-Román, S., & Salas Costumero, L. (2020). The Importance of Alexithymia in Post-surgery. Differences on Body Image and Psychological Adjustment in Breast Cancer Patients. Frontiers in psychology11, 604004. https://doi.org/10.3389/fpsyg.2020.604004

Response: Thank you very much for your comments. We have made these changes in the introduction of the manuscript

*Method

Mastectomies can be classified into several types. What specific procedure was performed on the patients in the study? I suggest explaining this in the demographic and clinical results of the sample.

Response: Thank you for your comment. We added this information in the manuscript (See lines 115-120)

It is not clear whether the reliability index of the instruments is the one drawn from this sample or whether it is the one drawn from the validation of the questionnaire.

Response: Thank you very much for pointing this out. The reliability of the instruments is taken from the literature. (See lines 148-150).

"Bjelland I, Dahl AA, Haug TT, Neckelmann D (2002) The validity of the Hospital Anxiety and Depression Scale An updated literature review. Journal of Psychosomatic Research 9". (See lines 148-150)

I do not understand the relationship between the Introduction and the Method. According to the statistical pathway, several groups are made according to age and other sociodemographic variables. However, this information does not appear in the previous theoretical framework. There is no connection of ideas.

Response: Thank you very much for your comments. The introduction has been reworded in order to be adapted to the intended purpose of the study.

*Results

They are well written and understandable. However, as I have previously indicated, they do not follow a sense in relation to the research question formulated nor to the title of this manuscript.

Response: Thank you for your comment. Se ha modificado el título y objetivo del estudio para adecuarlo a los resultados.

“Related factors with depression and anxiety in mastectomized women breast cancer survivors”

“The aim of this study was to identify factors associated with depression and anxiety in mastectomized women BCS”

*Discussion

Some methodological limitations are lacking in relation to the type of study. For example, a cross-sectional study does not allow confirmation of hypotheses or maintenance of results over time.

The Discussion section could benefit from a paragraph of clinical implications beyond a sentence indicating the allocation of resources in the Health System. Specifically, what resources do you want to make available to patients with this problem?

Response:  Thank you very much for pointing this out, we have added information regarding this in lines 341-350 and 359-367

We hope we have now answered all your comments and we are looking forward to hearing from you again.

Jessica Fernández Solana, OT, PT

Reviewer 2 Report

Inclose

Author Response

Mrs. Jessica Fernández Solana

Department of health sciences

University of Burgos, Paseo Comendadores s/n.

Burgos, 09001, Spain

Tel. (+34) 947499108

08-12-2022

IJERPH.  Subject: Submissions Needing Revision

Dear editor.

Thank you very much for inviting us to submit our response to reviewers for our manuscript (ijerph-2014157) entitled: “Quality of life in breast cancer survivors in relation to age, type of surgery and length of time since first treatment”

We have checked our manuscript according to the Academic Editor, the reviewers’ comments and the Journal requirements. We have also responded to some comments from reviewers point by point).

We would be very grateful if you could consider our manuscript to be published in your journal.

Yours sincerely,

Jessica Fernández Solana, OT, PT

  1. Response to Reviewer 2:

First of all, we would like to express our sincere gratitude for all comments and suggestions received from the Reviewer 2. This information has certainly enriched the text for its best understanding, thank you very much indeed. We have clarified the reviewer2’s questions. We have introduced the required changes both in our answers to the specific comments and in the final manuscript V2.

Line 20 – “being considered pathological” Use pathological between “ “ or use severe at this point

 Response: Thank you for your comment. Changes have been made to the manuscript

Line 22 - There is no use for psychotherapy in these patients. What kind of psychotherapy do you use? The idea, in this context, is to make psychological treatment, not psychotherapy

 Response: Thank you very much for pointing this out. This concept has been deleted.

Line 68 – “the psychiatric needs of their patients” – Psychiatric and psychological needs

 Response: Thank you for your comment, we have remove this section.

Line 83 – Counselling; interventions programmes, Psychological treatment, you name it. But I don't think so Psychotherapy will fit here

 Response: Thank you very much for pointing this out, we have added this information in the manuscript (See line 100).

Line 133 - not mental disorders. Depression and anxiety. Maybe you can say "...these mental disorders"

Response: Thank you very much for pointing this out, changes have been made.

Line 260 – “This may be because BCS who do have a family history may 260 have an example of what happened or may even receive advice from family members 261 who have already had the disease and know what it is like to deal with it.” You are speculating Line

 Response: Thank you for your comment. No bibliographical reference has been found to justify this. The commentary is based on work and research experience with this type of patient.

Line 268 – “This is because BCS are worried” Use probably

 Response: Thank you for your comment. Changes have been made.

Line 285 – “These data show the high prevalence of depression and anxiety in our sample compared to other populations” - Can you try to discuss why?

Response: Thank you for your comment, we have added this information in the manuscript (See lines 336-340)

Line 296 – Use psychological treatment You should correlate the results with patients who have psychological treatment. It would be important to do so.

 Response: Thank you for your comment, we have amended this issue (See lines 342)

It has not been possible to correlate the results with the patients who have psychological treatment since they were not taken into account in this study, but they will be taken into consideration for future research work, so they have been added in the limitations (See lines 341-343).

We hope we have now answered all your comments and we are looking forward to hearing from you again.

Jessica Fernández Solana, OT, PT

Reviewer 3 Report

---------------------------------------------------------------------

Manuscript ID: ijerph-2034843

Factors associated with depression and anxiety in mastectomised women breast cancer survivors

---------------------------------------------------------------------

The manuscript entitled “Factors associated with depression and anxiety in mastectomised women breast cancer survivors” tries to identify factors associated with depression and anxiety. The study presents a retrospective cross-sectional design, which includes data from mastectomized women survivors of breast cancer.

The topic is interesting and I believe it is worth further study. Breast cancer is a tough enough condition in its own right even for survivors, which may include related health problems, the stigma derived from the mastectomy, multiple medical follow-ups and restrictions on activities of daily living. If it is possible to identify factors related to patients' mental health and behavioral disorders, more appropriate interventions and prevention programs can be derived.

After reading the manuscript, I have some concerns about the statistical analyses that I think can be resolved. Here are some comments I would like to offer to the authors.

I do not think “mastectomised” is the correct spelling of the term. Not being an English speaking native I cannot be sure, but “mastectomized” sounds better.

The abstract does not contain information on which are the factors associated with depression and anxiety in the population (it only describes the prevalence of depression and anxiety in the sample).

Procedure

p. 3. “The evaluation of the participants was carried out at two points in time, for the first one, data was collected in two occasions. Firstly…” That expression suggests a second point in time, but it was never mentioned in de Procedure. Please clarify.

p. 3. The author classified the patients by age ranges (< 50 years. between 50 and 65 years and > 65 years). By chopping the variable, you are losing valuable information about age. This method reduces variance and bias the effect size estimates of significant tests. It makes no sense to do that (unless there is a theoretical criterion that justifies it), especially if the original, continuous variable is expressed in years of age. Please, use the age as a continuous variable.

The same goes to elapsed time. Why the 5 years threshold? Please use the continuous variable (again, unless there is a theoretical criterion to do otherwise).

Statistical analysis

p. 3. Were the assumptions of the statistical analyses met?  The results of the previous analyses (that should be reported) may change the significance tests to be used.

p. 4. “Differences in the total assessment of depression and anxiety between groups were tested using a univariate test with the group to which each patient belonged as the fixed factor”. I think this method is not appropriate (or I did not understand how it was conducted; was it a univariate t-test?). As we have relevant information for both groups (mean and SD), a bivariate test is recommended. Please revise and state clearly which test was used.

p. 4. Something is wrong with the wording of the sentence about effect sizes: “...where there is no effect if 0 2 p 154 < 0.05; minimal effect if 0.05 2 p < 0.26; moderate effect if 0.26 2 p < 0.64; and strong effect 155 if 2 p 0.64”. Please revise. Anyway, I do not think Ferguson (2009) proposed such thresholds for effect size. Firstly, Ferguson only displays in a table recommended minimum effect sizes, but they are not of his making. Secondly, partial eta squared estimates are used for factorial ANOVA designs, which is not the case in this study.

Moreover, whatever the effect size index we use, we should always try to interpret effect sizes in a situational context. The authors reported eta squared estimates, but they are not interpreted in any way. Please select adequate effect size estimates and interpret them in a meaningful way.

Discussion and Conclusions

The authors presented conclusions based on profiles of patients (e.g. “BCS under 50 years of age, who have received some form of treatment, with no family history, no partner, employed, with more than a high school education and with more than 5 years since diagnosis”), but the statistical analyses are not adequate to conclude so. A independent significance test for each factor is all right, as a standalone procedure, to study the relationship between two variables, but multiple independent test do not allow to elaborate profiles based on significant factors. It would be better to fit a multiple regression model, which would allow identifying the variables that explain the most variance of anxiety and depression.

Author Response

Mrs. Jessica Fernández Solana

Department of health sciences

University of Burgos, Paseo Comendadores s/n.

Burgos, 09001, Spain

Tel. (+34) 947499108

30-11-2022

IJERPH.  Subject: Submissions Needing Revision

Dear editor.

Thank you very much for inviting us to submit our response to reviewers for our manuscript (ijerph-2014157) entitled: “Quality of life in breast cancer survivors in relation to age, type of surgery and length of time since first treatment”

We have checked our manuscript according to the Academic Editor, the reviewers’ comments and the Journal requirements. We have also responded to some comments from reviewers point by point).

We would be very grateful if you could consider our manuscript to be published in your journal.

Yours sincerely,

Jessica Fernández Solana, OT, PT

  1. Response to Reviewer 3:

First of all, we would like to express our sincere gratitude for all comments and suggestions received from the Reviewer 2. This information has certainly enriched the text for its best understanding, thank you very much indeed. We have clarified the reviewer2’s questions. We have introduced the required changes both in our answers to the specific comments and in the final manuscript V2.

The manuscript entitled “Factors associated with depression and anxiety in mastectomised women breast cancer survivors” tries to identify factors associated with depression and anxiety. The study presents a retrospective cross-sectional design, which includes data from mastectomized women survivors of breast cancer.

The topic is interesting and I believe it is worth further study. Breast cancer is a tough enough condition in its own right even for survivors, which may include related health problems, the stigma derived from the mastectomy, multiple medical follow-ups and restrictions on activities of daily living. If it is possible to identify factors related to patients' mental health and behavioral disorders, more appropriate interventions and prevention programs can be derived.

After reading the manuscript, I have some concerns about the statistical analyses that I think can be resolved. Here are some comments I would like to offer to the authors.

I do not think “mastectomised” is the correct spelling of the term. Not being an English speaking native I cannot be sure, but “mastectomized” sounds better.

 Response: Thank you very much for pointing this out, we have modified this word in the manuscript.

The abstract does not contain information on which are the factors associated with depression and anxiety in the population (it only describes the prevalence of depression and anxiety in the sample).

 Response: Thank you very much for pointing this out, we have added this information in the manuscript (See lines 20-23).

Procedure

  1. 3. “The evaluation of the participants was carried out at two points in time, for the first one, data was collected in two occasions. Firstly…” That expression suggests a second point in time, but it was never mentioned in de Procedure. Please clarify.

 Response: Thank you very much for pointing this out, we have modified this section (See line 125)

  1. 3. The author classified the patients by age ranges (< 50 years. between 50 and 65 years and > 65 years). By chopping the variable, you are losing valuable information about age. This method reduces variance and bias the effect size estimates of significant tests. It makes no sense to do that (unless there is a theoretical criterion that justifies it), especially if the original, continuous variable is expressed in years of age. Please, use the age as a continuous variable.

 Response: Thank you very much for your comments. This choice has been justified on the basis of the existing literature (See lines 169-171).

The same goes to elapsed time. Why the 5 years threshold? Please use the continuous variable (again, unless there is a theoretical criterion to do otherwise).

 Response: Thank you very much for your comments. This choice has been justified on the basis of the existing literature (See lines 171-175).

Statistical analysis

  1. 3. Were the assumptions of the statistical analyses met?  The results of the previous analyses (that should be reported) may change the significance tests to be used.

 Response: Thank you for the comment, the analyses performed are indicated in the document. A new linear regression analysis has been added for each of the variables used.

  1. 4. “Differences in the total assessment of depression and anxiety between groups were tested using a univariate test with the group to which each patient belonged as the fixed factor”. I think this method is not appropriate (or I did not understand how it was conducted; was it a univariate t-test?). As we have relevant information for both groups (mean and SD), a bivariate test is recommended. Please revise and state clearly which test was used.

 Response: Thank you for the comment. sorry, it was expressed incorrectly. It has been amended to clarify that the analysis performed was a bivariate analysis, using a t-test for independent samples. A linear regression analysis has been added (See lines 180-182)

  1. 4. Something is wrong with the wording of the sentence about effect sizes: “...where there is no effect if 0 2 p 154 < 0.05; minimal effect if 0.05 2 p < 0.26; moderate effect if 0.26 2 p < 0.64; and strong effect 155 if 2 p 0.64”. Please revise. Anyway, I do not think Ferguson (2009) proposed such thresholds for effect size. Firstly, Ferguson only displays in a table recommended minimum effect sizes, but they are not of his making. Secondly, partial eta squared estimates are used for factorial ANOVA designs, which is not the case in this study.

 Response: Thank you for your comment. The sentence has been corrected and the bibliographic reference has been modified. The criteria of Cohen (1988) have been used. (See lines 184-186).

“Finally, statistical power and effect sizes were calculated. Effect sizes were deter-mined using Cohen´s criteria (partial eta squared (η2)), where there is a small effect if η2 < .059 (d= 0.20), the effect is medium when η2 ≥ 0.59 (d= 0.50); and the effect size is large when η2 ≥ .138 (d= 0.80)”

Moreover, whatever the effect size index we use, we should always try to interpret effect sizes in a situational context. The authors reported eta squared estimates, but they are not interpreted in any way. Please select adequate effect size estimates and interpret them in a meaningful way.

Response: Information has been added in the manuscript (See lines 220-223).

Discussion and Conclusions

The authors presented conclusions based on profiles of patients (e.g. “BCS under 50 years of age, who have received some form of treatment, with no family history, no partner, employed, with more than a high school education and with more than 5 years since diagnosis”), but the statistical analyses are not adequate to conclude so. A independent significance test for each factor is all right, as a standalone procedure, to study the relationship between two variables, but multiple independent test do not allow to elaborate profiles based on significant factors. It would be better to fit a multiple regression model, which would allow identifying the variables that explain the most variance of anxiety and depression.

 Response: Thank you for your comment. A linear regression analysis has been added, in which employment status significantly predicts depression and elapsed years significantly predicts anxiety (See lines 224-228).

We hope we have now answered all your comments and we are looking forward to hearing from you again.

Jessica Fernández Solana, OT, PT

Round 2

Author Response

Mrs. Jessica Fernández Solana

Department of health sciences

University of Burgos, Paseo Comendadores s/n.

Burgos, 09001, Spain

Tel. (+34) 947499108

29-12-2022

IJERPH.  Subject: Submissions Needing Revision

Dear editor.

Thank you very much for inviting us to submit our response to reviewers for our manuscript (ijerph-2034843) entitled: “Related factors with depression and anxiety in mastectomized women breast cancer survivors”

We have checked our manuscript according to the Academic Editor, the reviewers’ comments and the Journal requirements. We have also responded to some comments from reviewers point by point).

We would be very grateful if you could consider our manuscript to be published in your journal.

Yours sincerely,

Jessica Fernández Solana, OT, PT

  1. Response to Reviewer 3:

First of all, we would like to express our sincere gratitude for all comments and suggestions received from the Reviewer 3. This information has certainly enriched the text for its best understanding, thank you very much indeed. We have clarified the reviewer3’s questions. We have introduced the required changes both in our answers to the specific comments and in the final manuscript V3.

The authors have modified the manuscript, now entitled “Related factors with depression and anxiety in mastectomized women breast cancer survivors”. There are still some aspects that should be improved or clarified before it can be submitted for publication. I am concerned about the statistical soundness of the study, and therefore of the conclusions drawn from it. Some of my previous comments on this issue were not addressed, and other modifications posed new problems. I am afraid that a revision of the statistical methods is needed, and that the conclusions have to be carefully reworked. I add below some comments that may help future submissions 

Procedure

  1. 3. You have changed “The evaluation of the participants was carried out at two points in time, for the first one, data was collected in two occasions. Firstly…” for “The evaluation of the participants was carried out at two points in time. Firstly …”. The problem I mentioned in my first review partially remains. “Firstly” suggests a “secondly” clause. Please clarify.

Response: Thank you very much for pointing this out, we have added this information in the manuscript (See line 143) “Secondly an assessment was conducted”.

Although I would have used the numeric variable for age, I now understand your criterion of using 50 years as the threshold for dichotomizitation. However, the use of the 5 years cut-off for elapsed time is not sufficiently justified, especially since there is no consensus in the literature. Please use elapsed time as a numeric, continuous variable

Response: Thank you very much for pointing this out, we have added this information in the manuscript (See lines 177-180) “(<5 years vs. ≥ 5 years - this 5-year cut-off has been used, since doctors use this criteria to estimate that the cancer has been overcome, even aromatase inhibitor treatments and drugs such as tamoxifen are prescribed for a period of 5 years after the last chemotherapy or radiotherapy treatment [33])”

Statistical analysis

In my previous review I asked about the assumptions of the statistical analyses (whether t-tests of regression models), that should be reported in the manuscript. The answer has been “Thank you for the comment, the analyses performed are indicated in the document. A new linear regression analysis has been added for each of the variables used”. That does not address the issue.

 Response: Thank you very much for pointing this out, we have added this information in the manuscript (See lines 185-187) “Differences in the total assessment of depression and anxiety between groups were tested by bivariate analysis, performing a t-test for independent samples, with the group to which each patient belonged as a fixed factor”.

  1. 4 “Linear regression analysis was also performed”. The authors took my comments into account and included a regression model. I appreciate that. However, what type of regression model was used? Simple, multiple? Hierarchical? If so, forward method? Stepwise? Which were the criteria of variable inclusion? These details should be explained in this section.

Response: Thank you very much for pointing this out, we have added this information in the manuscript (See lines 187-191) “To analyse factors associated to depression and anxiety in BC mastectomized women, a stepwise regression was used with age, elapsed years, treatment, type of surgery, family history, marital status, employment status and level of education as independent variables and depression and anxiety scores as dependent variables”.

Statistical analysis and Results

Effect sizes: Firstly, citation [32] references an applied paper about the effects of type of surgery and time on psychological adjustment in BCS, and does not allow justifying those (or any) η2 effect size thresholds. Secondly, if my memory serves, the eta-squared thresholds are incorrect (and, anyway, it makes no sense that η2 ≥ 0.59 is “medium” and η2 ≥ .138 is “large”). Thirdly, how did you compute η2 for a bivariate t-test using SPSS? Lastly, Cohen himself (a different Cohen!) stated that “the proposed conventions will be found to be reasonable by reasonable people”. As the effect size indices are, at the very least, arbitrary, it is necessary to provide some context for interpreting the r2 values. In this version, the authors do not interpret eta-squared estimates, just gave them arbitrary labels that were not discussed further. Now I will repeat some of my previous comments on the subject: whatever the effect size index we use, we should always try to interpret effect sizes in a

situational context. Please select adequate effect size estimates and interpret them in a meaningful way. Perhaps the clinical interpretation of HADS values can be of helpful for this task. 

Response: Thank you very much for pointing this out, we have added this information in the manuscript (See lines 192-195 “statistical power and effect sizes were calculated by software SPSS. Effect sizes were determined using Ferguson's criteria (partial eta squared (η2)), where there is a small effect if η2 < .04, the effect is medium when η2 0.25; and the effect size is large when η2 .64” ; 226-230 “low effect size”, 233-235 “The effect size is low since the eta-squared does not reach the recommended minimum effect size representing a "practically" significant effect described by Ferguson [34], so we should interpret these results with caution” and 362-364 “In addition, another major limitation and one for which we should interpret these results with caution is the low side effect indicated by the them according to Ferguson's criteria”.

Results

  1. 5. The text below Table 1 repeats information already present in the table. This redundancy is not desirable.

Response: Thank you very much for pointing this out, we have modified this section (See lines 197-198)

  1. 5. “The effect size is small for all variables, although the power of the test is high”. What do you mean by that? I am concerned about this conceptual confusion. Observed power is not the same as effect size, and their estimators serve different purposes. A high observed power does not compensate in any case for a small effect size. But, anyway, a small effect size can be relevant in the right context.

Response: Thank you very much for your comments. We have removed this section.  

Discussion and Conclusions

The authors presented conclusions based on profiles of patients (e.g. “BCS under 50 years of age, who have received some form of treatment, with no family history, no partner, employed, with more than a high school education and with more than 5 years since diagnosis”), but the statistical analyses are not adequate to conclude so. This is my exact comment of the previous review. This time the authors added a regression model (may be a multiple regression model where all explanatory variables where introduced, although the details are missing, as stated above). Only two predictors were significant (employment status and years elapsed for depression and anxiety, respectively). How is it that the

conclusion is the same as before adding the regression model? This issue is of vital importance and requires a revision of the manuscript. 

Response: Thank you very much for your comments. Changes have been made in this section.

(See lines 26, 29, 369, 372 “could have”; 377-378 “Likewise, a small relationship has been observed between depression and employment situation, and anxiety with elapsed years”.

We hope we have now answered all your comments and we are looking forward to hearing from you again.

Jessica Fernández Solana, OT, PT
